# Deficits in Mitochondrial Spare Respiratory Capacity Contribute to the Neuropsychological Changes of Alzheimer’s Disease

**DOI:** 10.3390/jpm10020032

**Published:** 2020-04-29

**Authors:** Simon M. Bell, Matteo De Marco, Katy Barnes, Pamela J. Shaw, Laura Ferraiuolo, Daniel J. Blackburn, Heather Mortiboys, Annalena Venneri

**Affiliations:** Sheffield Institute for Translational Neuroscience (SITraN), University of Sheffield, 385a Glossop Road, Sheffield S10 2HQ, UK

**Keywords:** Alzheimer’s disease, mitochondrial spare respiratory capacity, mitochondrial, membrane potential, glycolytic reserve, semantic memory, phonemic fluency, episodic memory, neuropsychology, neuroimaging

## Abstract

Alzheimer’s disease (AD) is diagnosed using neuropsychological testing, supported by amyloid and tau biomarkers and neuroimaging abnormalities. The cause of neuropsychological changes is not clear since they do not correlate with biomarkers. This study investigated if changes in cellular metabolism in AD correlate with neuropsychological changes. Fibroblasts were taken from 10 AD patients and 10 controls. Metabolic assessment included measuring total cellular ATP, extracellular lactate, mitochondrial membrane potential (MMP), mitochondrial respiration and glycolytic function. All participants were assessed with neuropsychological testing and brain structural MRI. AD patients had significantly lower scores in delayed and immediate recall, semantic memory, phonemic fluency and Mini Mental State Examination (MMSE). AD patients also had significantly smaller left hippocampal, left parietal, right parietal and anterior medial prefrontal cortical grey matter volumes. Fibroblast MMP, mitochondrial spare respiratory capacity (MSRC), glycolytic reserve, and extracellular lactate were found to be lower in AD patients. MSRC/MMP correlated significantly with semantic memory, immediate and delayed episodic recall. Correlations between MSRC and delayed episodic recall remained significant after controlling for age, education and brain reserve. Grey matter volumes did not correlate with MRSC/MMP. AD fibroblast metabolic assessment may represent an emergent disease biomarker of AD.

## 1. Background

Alzheimer’s disease (AD) is the most common cause of dementia worldwide and in 2018 was estimated to cost the global economy 1 trillion US dollars [1]. The clinical symptoms of the disease are the progressive loss of different aspects of cognitive function until a patient becomes completely dependent on the care of family members and healthcare workers [2]. Median survival after diagnosis is 7 to 10 years for people in their 60s to 70s, and 3 years for people in their 90s [3]. 

The disease is characterized pathologically by the presence of extracellular amyloid plaques comprising mainly of the amyloid beta protein; and intracellular neurofibrillary tangles (NFT) made mainly of the cytoskeletal protein tau [4]. To date the cause of AD still remains poorly understood. As the buildup of amyloid appears to be a key step in the development of both familial and sporadic forms of the disease, the amyloid cascade hypothesis has become the leading theory for the cause of the condition [5]. In brief, this hypothesis states that the key step in developing AD is the accumulation of amyloid beta through reduced breakdown and clearance, and/or increased production. Strategies aimed at reducing the amyloid load in the brain, however, have failed to control the disease [6] and have resulted in a large number of clinical trials that have failed to achieve primary outcome measures [7]. Even in pre-clinical carriers of dominantly inherited AD mutations, amyloid removal therapies have not slowed disease progression [8]. Furthermore, brain amyloid load does not correlate with clinical symptoms [9]. This has led researchers to investigate alternative pathophysiological mechanisms [10].

AD is clinically defined by distinctive changes in cognitive status identified by neuropsychological assessment. Brain imaging changes and amyloid and tau protein levels in the cerebrospinal fluid are used to confirm the diagnosis in vivo [11]. The changes seen in a patient’s ability to perform a cognitive task are often difficult to explain from a cellular perspective. Tau deposition does explain elements of the observed neuropsychological abnormalities [12], but does not fully account for all cognitive changes observed in AD patients [13,14].

Cognitive processing, such as that required while performing memory tasks, puts an increased metabolic demand on the brain [15]. This is evidenced by neuroimaging studies of the brain which use tracers of metabolism such as 2-[18F]fluoro-2-Deoxy-D-glucose (FDG) that have shown poor glucose utilization in patients who perform poorly on memory tasks [16,17]. Positron-emission tomography (PET) imaging studies that use oxygen-15 labelled water also show reduced uptake when AD patients perform cognitive tasks [18]. These imaging studies suggest that any deficit in metabolic function, such as deficits in mitochondrial respiration or glycolysis, are likely to affect an individual’s performance on cognitive tasks. It is therefore possible that mitochondrial respiration or glycolytic dysfunction might contribute to cognitive deficits in AD, making these cellular processes suitable pharmacological targets to improve the cognitive symptoms of AD.

Cellular metabolic changes within the brains of patients with AD and in peripheral cell populations are seen very early in the condition, and often precede the development of both amyloid plaques and NFT. Abnormalities have been shown in many metabolic pathways in AD [19]. Mounting evidence suggests that deficits in glycolysis and the function of mitochondria, specifically how they control oxidative phosphorylation, are likely to be key in the development and establishment of AD [20,21,22].

A mitochondrial cascade hypothesis has been suggested for the aetiology of AD, and states that people who inherit mitochondrial genes that predispose them to lower mitochondrial respiration rates may be more likely to develop the condition [23]. In animal models of AD, changes in mitochondrial function are seen prior to amyloid deposition [20,24], and cell models show changes in mitochondrial function and oxidative stress without the presence of amyloid [25], giving further evidence for the key role of mitochondrial dysfunction in AD.

Impairment of glycolysis is also seen early in patients with AD. A 2-[18F]fluoro-2-Deoxy-D-glucose positron-emission tomography (FDG-PET) imaging of the brain shows a reduction in glucose metabolism [26]. In particular, there is a reduction in aerobic glycolysis in brain areas susceptible to amyloid deposition [27], and in those regions where high levels of tau accumulation are seen [28].

We have previously shown that fibroblasts from sporadic AD (sAD) patients have multiple mitochondrial structural and functional abnormalities, and that these can be ameliorated by treatment with ursodeoxycholic acid (UDCA) [29]. Other studies have shown that glycyl-l-histidyl-l-lysine (GHK-Cu), by increasing gene expression, has an effect on improving mitochondrial activity and influencing cognitive decline [30]. In our previous study we showed sAD fibroblasts to have deficits in mitochondrial membrane potential (MMP) and mitochondrial spare respiratory capacity (MSRC). MSRC refers to the difference in oxidative phosphorylation rates between the basal level of mitochondrial respiration and the maximal level a cell can achieve [31]. In essence, MSRC measures the cellular reserve respiratory capacity. In animal models of AD it has been shown that deficits in MSRC cause cognitive deficits that treatment with the antioxidant pyrroloquinoline quinone can improve [32]. It has not been previously shown, however, whether changes in MSRC in human cell lines obtained from sAD patients correlate with their performance on neuropsychological tests.

As deficits in both glycolysis and mitochondrial function have been shown in AD patients and models, in this proof of concept study we explored metabolic function in fibroblasts from sporadic AD patients and its role in the cognitive decline experienced by these patients. To this end, we assessed mitochondrial functional changes in a larger cohort of patients compared to the findings described in our previous study [29]. We have then assessed additional metabolic parameters in the sAD fibroblasts including glycolytic function and cellular ATP levels, which have previously not been described. Finally, we investigated whether these metabolic abnormalities detected in sAD fibroblasts correlated with neuropsychological and neuroimaging features typical of the early stages of this disease.

## 2. Results

### 2.1. Patient Demographic Details

Skin biopsies were taken from ten sAD patients (mean age 61.3 years, 6 male) and ten controls (mean age 66.7 years, 5 male). Body mass index (BMI) did not differ significantly between the groups (sAD mean 27.8 kg/m^2^ SD 5.37 kg/m^2^ Controls mean 28.2 kg/m^2^ SD 4.00 kg/m^2^
*t*-test *p =* 0.44). Table 1 shows patient and control demographic data.

### 2.2. Neuropsychological Profiles

Neuropsychological profiling of controls and patients used in this study was performed as part of the VPH-DARE research project (http://www.vph-dare.eu/), (see Methods section for additional details). For the present study, a subgroup of neuropsychological tests was selected to be correlated with metabolic findings. The tests which can detect the earliest typical cognitive impairments in mild sAD were selected. These included assessment of semantic memory, immediate and delayed episodic recall. Phonemic fluency was also selected as dysfunction on this cognitive test is seen later in AD but not in its earlier stages [33]. All 10 controls and 10 patients with AD were assessed. The Mini-Mental State Examination (MMSE) [34] was also used, as this test is useful in staging disease severity in sAD [34,35].

Patients with sAD performed at a lower level than controls on all tests (see Figure 1). The most significant differences were seen in the tests of semantic memory (mean score in controls 42.6 points, mean score in sAD 17.5 points, difference between means 25.1 points, SD 4.774 *p <* 0.0001); immediate (mean score in controls 15.8 points, mean score in sAD 6.4 points, difference between means 25.1 points, SD 1.468 *p* < 0.0001); and delayed recall of the Prose Memory test (mean score in controls 18.6 points, mean score in sAD 5.5 points, difference between means 13.4 points, SD 1.488 *p <* 0.0001). The phonemic fluency test performance was also significantly different between the 2 groups (mean score in controls 50.5 points, mean score in sAD 27.4 points, difference between means 23.1 points, SD 6.837 *p =* 0.0033), but to a lesser extent than the other neuropsychological tests. As expected, MMSE scores of sAD patients were significantly lower than those of controls (mean score in controls 27.3 points, mean score in sAD 21.3 points, mean difference 6.0 points, SD 1.54 *p =* 0.0011).

### 2.3. Neuroimaging Profiles

Volumetric structural MRI scans were acquired on 9 controls and 10 patients with sAD. One control did not complete their MRI scan as an incidental finding (of no diagnostic significance for this study) on initial scans meant the participant could no longer take part in the VPH-DARE@IT original study. For the remaining participants, the left and right parietal lobes, anterior medial pre-frontal cortex and posterior cingulate gyrus, and left hippocampal grey matter volumes were extracted as these brain areas are affected early in AD [36].

Comparisons between the 2 groups showed no significant differences between the grey matter volumes of the preselected areas when performing a *t*-test. Brain grey matter volumes can be affected by age, education and brain reserve [37,38]. For these reasons a further analysis of the 2 groups was completed controlling for these covariates. A significant difference emerged in the volume of the left hippocampus, left parietal, right parietal and anterior medial prefrontal cortex in patients with sAD after controlling for confounding variables. Table 2 highlights the F-test and *p*-values for these differences. No significant difference was seen in the posterior cingulate cortex grey matter volume between the 2 groups.

### 2.4. Fibroblast Metabolic Assessment

Mitochondrial function was investigated in both sAD patients and controls. We measured elements of both mitochondrial function and glycolysis. Five metabolic parameters were chosen that best describe different factors influencing the ability of the fibroblast to meet its energy demands. Mitochondrial parameters included: total cellular ATP, as this is a global marker of the energetic status of the cell; MMP and MSRC. MMP and MSRC were selected as these parameters are important in maintaining the rate of ATP production as cellular energy demand changes. These are also the two parameters which we have previously identified as being relevant to the mitochondrial phenotype in sAD fibroblasts [29]. Measures of glycolysis included glycolytic reserve and extracellular lactate levels. Glycolytic reserve, like MSRC and MMP for mitochondrial function, is a measure of cellular metabolic flexibility and deficits in this parameter are likely to contribute to an inability to maintain cellular energy production. Extracellular lactate is the final breakdown product of glycolysis and contributes information about how well the cell can utilize glucose as a metabolite. 

First, we assessed MMP and MRSC as we have done in our previous work [29]. In this cohort, MRSC was significantly reduced (36% reduction *p <* 0.0001) in sAD when compared to controls (Figure 2A). MMP was also significantly reduced in the sAD fibroblast lines (14% reduction *p* = 0.011) (See Figure 2B). Figure 2C shows a representative oxygen consumption trace for the mitochondrial stress test experiment.

Next, we assessed total cellular ATP, which showed no significant difference between sAD patient fibroblasts and controls when comparing mean values (*p* = 0.165, Figure 2D). The final metabolic assessment of the fibroblast lines involved assessment of glucose metabolism using the glycolysis stress test programme on the Seahorse analyzer and extracellular lactate measurement. A significantly lower glycolytic reserve was found in sAD fibroblasts when compared to controls (see Figure 1E, 25.8% reduction, *p =* 0.031). No significant differences were seen in maximum glycolytic rate (*p =* 0.792), or basal glycolytic rate (*p =* 0.381). Figure 1F shows the extracellular acidification rate (ECAR) trace for the glycolysis stress test, comparing the group of controls against the sAD fibroblast group. Measurement of extracellular lactate levels revealed significantly lower lactate levels released from sAD fibroblasts (14% reduction, *p =* 0.0227, Figure 2G). 

### 2.5. Neuropsychological/Metabolic Correlations

Next, we sought to correlate the neuropsychological scores of the sAD patients and controls combined with their metabolic parameters measured in fibroblasts. As five neuropsychological measures and five fibroblast metabolic markers had been assessed to identify differences in the control and sAD fibroblast groups, we controlled for the effect of multiple comparisons by adjusting what we deemed to be a significant association to *p* ≤ 0.01 (0.05/5). Using this new statistical threshold, only MSRC and MMP metabolic tests and immediate, delayed and semantic memory neuropsychological assessments were identified as significantly different between control and sAD groups. Correlations were, therefore, performed only for these data sets.

MSRC had a significant positive correlation with immediate episodic recall (r = 0.612, *p =* 0.004), delayed episodic recall (*r =* 0.669, *p =* 0.001) and semantic memory scores (*r =* 0.614, *p* = 0.003). MMP correlated only with semantic memory scores (*r =* 0.565, *p =* 0.009). Immediate episodic recall (*r =* 0.552, *p =* 0.0134) and delayed episodic recall correlated positively with MMP, but at a lower level of significance (*r =* 0.540, *p =* 0.015). Figure 3 displays these correlations graphically.

Age, years of education and brain reserve can potentially confound the correlations between neuropsychological measurements and the fibroblast metabolic markers, as all three factors have been shown to affect either performance on neuropsychological tests [39,40] or mitochondrial function [41]. To control for the effect of these three covariates, a further analysis was performed taking these parameters into account. This further analysis showed that the correlation between delayed episodic recall and MSRC was still significant at the more conservative significance threshold. Both immediate episodic recall (*p =* 0.013) and semantic memory (*p =* 0.012) correlations with MSRC were no longer significant once covariates were included in the analysis at the new more stringent *p*-value. All correlations seen between the neuropsychological tests and MMP were no longer significant after controlling for age, education and brain reserve. Table 3 displays the new correlation coefficients and *p*-values for the neuropsychological data.

### 2.6. Neuroimaging/Metabolic Correlations

Volumetric imaging for areas of the brain known to be affected by AD were correlated with metabolic markers of the disease. Only the grey matter volumes that had shown significant differences between the sAD and control groups (with a *p*-value equal to or less than 0.01) were included. None of the grey matter volumes showed a significant correlation with changes in cellular metabolism. Table 4 shows the results of the correlations.

## 3. Discussion

This proof of concept study shows that fibroblast MSRC correlates with established neuropsychological abnormalities that are affected early in AD. This is one of the first studies to show a functional biomarker of AD correlating with a marker of fibroblast mitochondrial function. MSRC is a measure of the ability of mitochondria within cells to increase the production of ATP in response to an increased energy demand. The correlation of MSRC with neuropsychological scores potentially helps to describe the cellular pathology underlying these neuropsychological changes seen in AD. This is the first study in humans to show that peripheral cell MSRC correlates with neuropsychological profiles. Correcting the abnormality in MSRC could be a future therapeutic approach in AD. The fact that this abnormality in mitochondrial function correlates with a clinical biomarker of AD also opens the avenue for monitoring drug response in future clinical trials.

MSRC has been shown to be abnormal in multiple diseases including acute myeloid leukaemia (AML) [42], Parkinson’s disease [43,44] and motor neuron disease (MND) [45]. MSRC deficits may not be disease specific, but the combination of cellular metabolic changes seen here may represent a cellular metabolic profile specific to sAD. This metabolic phenotype could potentially represent a biomarker that can stratify and define a specific subset of sAD patients that might then respond to a personalised therapeutic approach. Previous research has shown that the metabolic profile identified in fibroblasts from sporadic Parkinson’s disease patients differs from that identified here in fibroblasts from sAD [46,47].

Reductions in MSRC and MMP were seen in all sAD fibroblast lines when compared to controls at a group level. These data reinforce and extend the findings reported in our previous paper [29], and show that the finding of reduced MSRC and MMP is reproducible and more robust in sporadic AD fibroblasts when sample size is increased. Extracellular lactate levels and cellular glycolytic reserves, both markers of glycolytic function, were significantly reduced in all sAD fibroblasts. These measures, however, did not meet the statistical significance for correlation with neuropsychological markers of the disease. The deficits in markers of mitochondrial function and glycolysis in the fibroblasts reflect a lack of flexibility of metabolic pathways in sAD. This lack of flexibility is likely to be very important when performing cognitive tasks that depend on the coordination of multiple brain regions. Taken together, these results suggest that sAD fibroblasts may have limited ability to respond to increased cellular energy demand.

MMP did correlate with neuropsychological markers, but correlations did not survive the correction for confounding factors unlike MSRC, for which correlation with delayed episodic recall (a core feature in the clinical diagnostic criteria for sAD) remained significant. Semantic memory and immediate episodic memory did not significantly correlate with MSRC at the more conservative significance threshold after confounder consideration, but *p*-values for both of these correlations were less than 0.02. Potentially, these correlations would reach significance level with an increased sample size. The difference in MMP between control and sAD groups was much smaller than the difference seen in MSRC. MMP has a key role in maintaining many pathways in the mitochondria outside that of ATP generation, such as apoptosis fates [48]. It is likely that several mechanisms not affected by sAD in the fibroblast cell help to maintain MMP levels, which may explain the reduced variability seen in this parameter in sAD and the lack of a significant correlation with neuropsychological scores. 

Baseline ATP levels were not significantly different between the two groups. These results suggest that, in a controlled environment when cells are not stressed, they can maintain ATP levels similar to that of controls. Interestingly though, the functional capacity of mitochondria is impaired, as described above, suggesting a potential inability to respond appropriately to increased ATP demand. The measurements of ATP we performed in this study only gives information about the total cellular ATP level. We did not investigate the rate of ATP breakdown nor how ATP levels change when the cell is under stressed conditions or the ratio of ATP/ADP in the cell. This may explain why no significant correlations were seen with ATP measurements, but were seen with the other markers of mitochondrial respiration. 

It is interesting that the correlation between scores on phonemic fluency, a neuropsychological test not affected early in sAD, and cell metabolism markers did not reach significance threshold. It. The medial temporal lobe is important in semantic memory processing [49], an area of cognition that is impaired very early in sAD [50]. This is not the case for phonemic fluency which is supported by processes associated with frontal executive regions [51]. It could be that the correlations we see between neuropsychological tests and cellular metabolic function may reflect which areas of the brain are more susceptible to metabolic failure. It has been previously shown that areas of the brain that are affected early in sAD express lower levels of electron transport chain (ETC) genes when compared to controls [52], suggesting that this might be the case. Data from FDG-PET studies also support the concept of focal brain hypometabolism in sAD, with areas such as the medial temporal lobes, precuneus and lateral parietal lobes all preferentially affected [53].

We did not see a significant correlation between mitochondrial function or glycolysis and brain structural imaging markers of sAD. This may be explained by the fact that multiple factors can affect grey matter volume such as age, levels of education and brain reserve. It has been previously shown that smaller grey matter volumes in the frontal lobes associated with ageing are associated with worse performance on frontal lobe cognitive tests [54]. The neuropsychological assessment performed in this study mainly focused on temporal lobe cognitive functioning and not frontal lobe function. Potentially, the effect of brain ageing on grey matter volumes may mask any effect metabolic function may have.

The limitations of our study include a small sample size, and the fact that we have used fibroblasts to measure metabolic function. It could also be argued that a correlation between metabolic function of central nervous system cells and neuropsychological parameters would be more meaningful. This is not the first study, however, to show that the metabolism of peripheral cells outside the nervous system is affected in sAD. White blood cells [55], platelets [56] and fibroblasts have been shown to have metabolic abnormalities in multiple studies [46,47,57,58]. Identifying that fibroblast mitochondrial abnormalities correlate with neuropsychological markers of AD opens up avenues to use fibroblast metabolic parameters as biomarkers of sAD, and also as a high throughput drug screening model, as well as potential outcome measures of therapeutic efficacy. Replication of the findings of this study in larger cohorts and by other groups is needed, however, to consolidate the evidence that fibroblast metabolic function is a reliable biomarker of sAD. 

Our patient cohort were selected at an early stage of sAD to try and reduce group variability. To understand the use of fibroblast metabolic abnormalities as a biomarker, further work investigating cohorts of patients with amnestic mild cognitive impairment (MCI), the prodromal stage of sAD, would be advantageous as well as investigating these parameters in more advanced sAD patients.

Future work could extend the findings of metabolic-neuropsychological correlations by creating neuronal lineage cells via cellular reprogramming methods such as that described by Takahashi et al. [59]. This type of model would allow for direct comparison between human nervous system cells and human nervous system function in a context where reprogrammed cells would maintain their ageing phenotype [60].

## 4. Methods 

### 4.1. Patient Details

Skin biopsies and fibroblast metabolic assessments were performed as part of the MODEL-AD research study (Yorkshire and Humber Research and Ethics Committee number: 16/YH/0155). Due to the initial success of fibroblast metabolic assessment, the initial cohort of four controls and four patients (previously reported in [29]) with sAD was expanded to ten healthy controls (mean age: 65.77 years, SD 14.70 years) and 10 patients with sAD (mean age: 61.33 years, SD 7.19 years). All patients had been involved in the EU-funded Framework Programme 7 Virtual Physiological Human: Dementia Research Enabled by IT (VPH-DARE@IT) initiative (http://www.vph-dare.eu/). A diagnosis of sAD was made based on clinical criteria [11]. Ethical approval for the neuropsychological and neuroimaging measures collected was gained from Yorkshire and Humber Regional Ethics Committee, Reference number: 12/YH/0474. Informed written consent was obtained from each participant. Investigations were carried out following the rules of the Declaration of Helsinki of 1975 (https://www.wma.net/what-we-do/medical-ethics/declaration-of-helsinki/), revised in 2013.

### 4.2. Neuropsychological Testing

A battery of tests was devised to detect impaired performance in the cognitive domains most susceptible to sAD neurodegeneration. This included tests of short- and long-term memory, attention and executive functioning, language and semantics, and visuoconstructive skills. A detailed description of each task is provided elsewhere [61]. Most tests were exclusively used as part of the clinical assessment of patients and controls. Three tests were also included as part of the experimental protocol. Immediate and delayed recall on the Logical Memory test were used as measures of episodic memory, the cognitive domain centrally defined by diagnostic criteria for a diagnosis of sAD. The performance on the Category Fluency test was used as an index of semantic memory, the cognitive domain affected by the accumulation of neurofibrillary pathology in the transentorhinal cortex [50]. Finally, the Letter Fluency test (phonemic fluency) was used as a methodological control, since performance on this task is often within normal age limits in patients with mild sAD [33,62].

### 4.3. MRI Acquisition and Processing

An MRI protocol of anatomical scans was acquired on a Philips Ingenia 3 T scanner. Several acquisitions (including T1-weighted, T2-weighted, FLAIR and diffusion-weighted sequences) were used for diagnostic purposes. Three-dimensional T1-weighted images (voxel size: 0.94 mm × 0.94 mm × 1.00 mm; repetition time: 8.2 s; echo delay time: 3.8 s; field of view: 256 mm; matrix size: 256 × 256 × 170) were also used for the calculation of hippocampal volumes. These were processed with the Similarity and Truth Estimation for Propagated Segmentations routine [63]. This procedure, available at niftyweb.cs.ucl.ac.uk, allows automated segmentation of the left and right hippocampus in the brain’s native space using multiple reference templates. Hippocampal volumes were then quantified using Matlab (version R2014a) and the “get totals” script (http://www0.cs.ucl.ac.uk/staff/G.Ridgway/vbm/get_totals.m). Fractional measures were also obtained dividing hippocampal volumes by the volume of the total intracranial space. The left hippocampal volume and left hippocampal ratio were chosen as structural markers of AD as evidence suggests the left hippocampus is affected early in the progression of the disease [64]. Native space maps created to extract hippocampal volume from MRI images are shown in Appendix A. Additional regions of interest (listed in Table 2) were then defined as binary image masks. Segmented grey matter maps were registered to the Montreal Neurological Institute space, and mask-constrained volumes were extracted.

### 4.4. Tissue Culture

Fibroblasts were cultured as described previously [29]. In brief, all 10 control and all 10 sAD lines were cultured in EMEM-based media (Corning) incubated at 37 °C in a 5% carbon dioxide atmosphere. Sodium pyruvate (1%) (Sigma Aldrich), non-essential amino acids (1%) (Lonza), penicillin and streptomycin (1%) (Sigma Aldrich), multi-vitamins (1%) (Lonza), Fetal Bovine Serum (10%) (Biosera) and 50µg/mL uridine (Sigma Aldrich) were added to the media. All experiments described were performed on all 20 cell lines. Fibroblasts were plated at a density of 5000 cells per well in a white 96 well plate for ATP assays. For MMP assays fibroblasts were plated at a density of 2500 cells per well in a black 96 well plate. Each assay was performed on three separate passages of fibroblasts; cells between passages 5–10 were used. All experiments were performed on passage-matched cells.

### 4.5. Intracellular ATP levels

Cellular adenosine triphosphate (ATP) levels were measured using the ATPlite kit (Perkin Elmer) as described previously [65]. ATP levels were corrected for cell number by using CyQuant (ThermoFisher) measurements, as previously described [29]. These values were then normalised to control levels.

### 4.6. Mitochondrial Membrane Potential

MMP was measured using tetramethlyrhodamine (TMRM) staining of live fibroblasts, as previously described [29]. Cells were plated in a black 96 well plate, incubated at 37 °C for 48 h. TMRM dye was added one hour prior to imaging on an InCell Analyzer 2000 high-content imager (GE Healthcare).

### 4.7. Extracellular Lactate Measurement

Extracellular lactate was measured from confluent flasks of fibroblasts using an L-Lactate assay kit (Abcam ab65331). The assay was performed according to the manufacturer’s instructions. 

### 4.8. Metabolic Flux Assay

#### 4.8.1. Mito Stress Test

Oxygen consumption rates (OCR) were measured using a 24-well Agilent Seahorse XF analyzer as described previously [29]. In brief, cells were plated at a density of 65,000 cells per well 48 h prior to measurement. Three measurements were taken at the basal point: after the addition of oligomycin (0.5 µM), after the addition of carbonyl cyanide-4-(trifluoromethoxy)phenylhydrazone (FCCP) (0.5 µM) and after the addition of rotenone (1 µM). OCR measurements were normalised to cell count as described in [29]. Measurements of basal mitochondrial respiration, maximal mitochondrial respiration and MSRC were also calculated.

#### 4.8.2. Glycolysis Stress Test

Cells were plated in a Seahorse XF24 Cell Culture Microplate (Agilent) at a density of 65,000 cells per well 48 h prior to measurement. The glycolysis stress test standard protocol was used [66]. This is a completely separate assay to the mitochondrial stress test, and allows for the thorough interrogation of glycolysis via the addition of supplements and inhibitors of the glycolytic pathway. Three measurements were taken at the basal point: after the addition of glucose (10 µM), after the addition of oligomycin (1.0 µM) and after the addition of 2-deoxyglucose (50 µM). The glycolysis stress test can measure several aspects of cellular glycolysis. These include the basal glycolytic rate of the cell, the maximum level of glycolysis the cell can achieve and the glycolytic reserve which refers to the difference between maximum level of glycolysis and the basal level of glycolysis. The different aspects of glycolysis are calculated by measuring the extracellular acidification rate (ECAR). ECAR rates were normalised to cell count as described above.

### 4.9. Statistical Analysis

For comparing each neuropsychological, neuroimaging and metabolic functional markers, a Student’s *t*-test was used, comparing the means of the control group for each parameter to the disease group. Statistics were calculated using GraphPad Prism Software (V7.02). Analyses of covariance were also used for group comparisons including confounding variables. For covariate analysis when comparing control and disease group grey matter volumes, or when correlating neuropsychological and neuroimaging sAD markers with fibroblast functional markers, the IBM SPSS Statistics suite (Version 26; https://www.ibm.com/uk-en/products/spss-statistics) was used as this function was not available in the GraphPad Prism Software. Significance levels were adjusted to account for multiple comparisons. 

## 5. Conclusions

These data highlight how in-depth analysis of mitochondrial and glycolytic function in sAD fibroblasts identifies metabolic abnormalities that parallel changes seen in neuropsychological features distinctive of the early stages of sporadic Alzheimer’s disease. This model system could be used to develop biomarkers useful in early detection as well as in the development of novel therapeutic approaches for sAD.

## Figures and Tables

**Figure 1 jpm-10-00032-f001:**
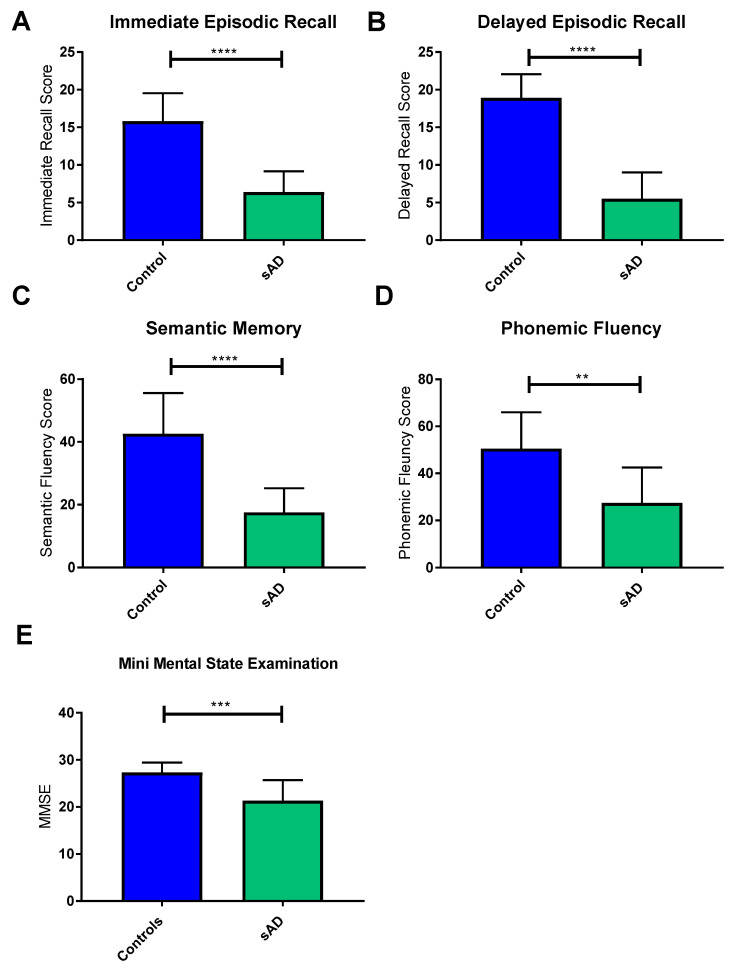
Mean scores of sporadic AD (sAD) patients and controls on cognitive tests included in the neuropsychological assessment. Graphs show mean with error bars indicating standard deviation. **** *p* < 0.0001; *** *p* < 0.001; ** *p* < 0.01. Controls are indicated by blue bars and sAD patients indicated by green bars. Significant reductions were seen in sAD immediate recall (**A**), delayed recall (**B**), semantic memory (**C**), phonemic fluency (**D**) and Mini Mental State Examination (**E**) when compared to controls.

**Figure 2 jpm-10-00032-f002:**
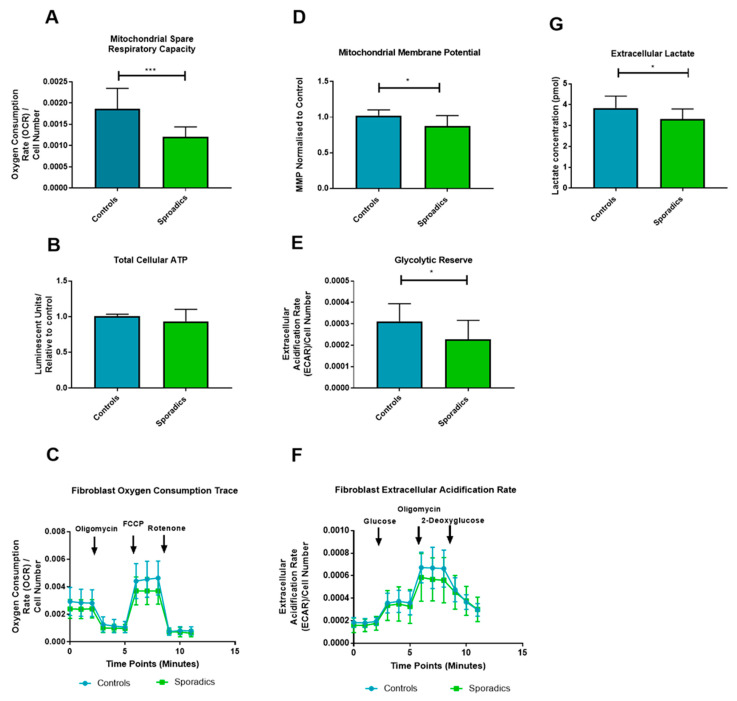
Oxidative phosphorylation and Glycolytic fibroblast Assessment: *** = *p <* 0.001; * = *p <* 0.05. Significant reductions in mitochondrial spare respiratory capacity (**A**), Mitochondrial Membrane potential (**D**), Glycolytic Reserve (**E**) and Extracellular lactate (**G**) was detected in sAD when compared to controls. No significant difference was seen in total cellular ATP (**B**). OCR and ECAR traces for mitochondrial stress test and glycolysis stress test are displayed in panels (**C**) and (**F**) respectively. For all panels controls are represented in blue and sAD are represented in green. Graphs represent mean with error bars indicating standard deviation.

**Figure 3 jpm-10-00032-f003:**
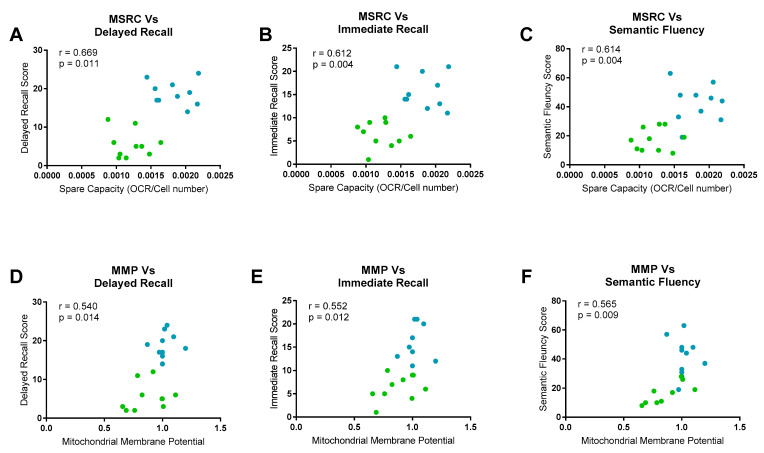
Neuropsychological Fibroblast Metabolism Correlations. Significant correlations were seen between mitochondrial spare respiratory capacity (MSRC) and delayed recall (**A**), Immediate recall (**B**) and semantic fluency (**C**). Mitochondrial membrane potential (MMP) correlated significantly with semantic fluency (**F**), and with delayed recall (**D**) and immediate recall (**E**) but to a lesser extent. For all panels controls are represented in blue and sAD are represented in green. Correlation coefficients and *p*-values for each correlation are displayed.

**Table 1 jpm-10-00032-t001:** Patient Demographic Information and contemporary treatment status.

Patient Number	Age (Years)	Sex	MMSE	Length of Education (Years)	AD Treatment (in Disease Cohort, at Time of Biopsy)
1	63	Male	20	16	None
2	57	Male	18	15	Donepezil
3	53	Male	14	11	Donepezil
4 *	60	Male	18	9	None
5 *	59	Female	23	11	Galantamine
6 *	63	Female	26	10	Donepezil
7 *	60	Male	18	11	Memantine
8	60	Male	18	11	None
9	79	Female	28	15	None
10	61	Female	25	11	Donepezil
Group Mean (Standard Dev)	61.33 (7.19)		20.8 (4.4)	12.0 (2.4)	
Controls					
1	66	Male	29	11	NA
2 *	54	Male	27	17	NA
3 *	53	Male	29	16	NA
4 *	56	Male	24	11	NA
5	61	Female	30	NA	NA
6	54	Female	29	17	NA
7 *	100	Female	24 #	14	NA
8	75	Female	28	18	NA
9	73	Female	26	12	NA
10	75	Male	27	10	NA
Group Mean (Standard Dev)	65.77 (14.7)		27.6 (1.8)	14 (3.1)	

# For control 7 some items of the Mini Mental State Examination (MMSE) could not be tested due to sensory impairment * indicates fibroblast lines in which Mitochondrial Membrane Potential (MMP) and Mitochondrial Spare Respiratory Capacity (MRSC) values were published in our previous paper [29].

**Table 2 jpm-10-00032-t002:** Differences in Control and sAD brain volumes. This table shows the significance of the differences in brain volumes between controls and patients with sAD when controlling for age, years of education and brain reserve.

Brain Volume	*F*-Test	*p*-Value
Left Hippocampal Volume	9.420	0.001
Left Parietal Volume	7.882	0.002
Right Parietal Volume	10.051	<0.0001
Anterior Medial Prefrontal Cortex	0.056	0.017
Posterior Cingulate Cortex	0.752	0.575

**Table 3 jpm-10-00032-t003:** Neuropsychological fibroblast metabolism correlations corrected for age, years of education and grey matter reserve.

Psychological Test	Fibroblast Marker	R Value	*p*-Value
Immediate Episodic Recall			
	MSRC	0.605	0.013
	MMP	0.196	0.466
Delayed Episodic Recall			
	MSRC	0.695	0.003
	MMP	0.204	0.448
Semantic Memory			
	MSRC	0.610	0.012
	MMP	0.345	0.191

**Table 4 jpm-10-00032-t004:** Grey Matter Volume Fibroblast metabolic correlations: No significant correlations were seen between MMP nor MSRC and any grey matter volume when controlling for age, length of disease or brain reserve.

Grey Matter Volume	Fibroblast Marker	R Value	*p*-Value
Left Hippocampal Volume			
	MSRC	0.371	0.157
	MMP	0.041	0.881
Left Parietal Volume			
	MSRC	0.341	0.196
	MMP	−0.047	0.862
Right Parietal Volume			
	MSRC	0.418	0.107
	MMP	−0.043	0.875

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
