# Peer review of "Deficits in Mitochondrial Spare Respiratory Capacity Contribute to the Neuropsychological Changes of Alzheimer’s Disease"

_jpm, 2020, doi:10.3390/jpm10020032_

Round 1

Reviewer 1 Report

This is a very  interesting paper in which generates data that has considerable potential in the the early diagnosis of AD and for developing targeted therapies, an issue that is obviously of huge clinical relevance. I am very much in support of the authors aims.

My concern with the paper is that the authors make huge interpretative demands of a sample that is very small for the type of analysis they undertake-namely regression analysis. I did not find evidence of Power calculations for the study. I am pleased that they undertook adjustments for the significance levels.

Throughout, the authors present data they refer to as correlation data but report r-squared rather than r-values. They also do not make it clear whether they are presenting adjusted r-squared values.  In addition Figure 3 graphs have what I would regard as regression lines-this is not appropriate for correlation analysis.  I would state that the authors need to clearly distinguish when correlation or regression analysis is being undertaken. I am satisfied that the authors undertake covariate adjustment, but this is a very risky strategy for samples that are as small as these and should be undertaken in a very limited manner.

The authors should not state "no correlation" but "no significant correlation"

The data presented is very interesting, researching AD in any of its forms is very challenging, and large samples are often difficult to obtain for studies such as this. I would ask that the authors report this as a proof of concept or pilot study or stress in more detail their limitations 

Author Response

Response to Reviewer 1

We thank the Reviewer for their comments and have listed below how we have addressed each point made.

My concern with the paper is that the authors make huge interpretative demands of a sample that is very small for the type of analysis they undertake-namely regression analysis. I did not find evidence of Power calculations for the study. I am pleased that they undertook adjustments for the significance levels.

We thank the Reviewer for their comment. To address the point they make we have referred to the study as a proof-of-concept study throughout the manuscript. This is to highlight the preliminary phase we are at in characterising the “mitochondrial phenotype” of AD and how this links to clinical variables. Studies like this which show positive findings in spite of a relatively small sample size are essential to encourage the design of larger studies which require more resources in terms of funding and commitment. On this note, we recognise that it is important to interpret our findings accordingly. For this reason, we have removed any direct or indirect reference to the application of regression models (see also response to the subsequent comment). In the limitations paragraph of the discussion (starts at line 370) we have also highlighted the need for both bigger cohorts of patients and replication of these findings by other groups before exploring fibroblast metabolism as an Alzheimer’s disease biomarker. This is also the reason we had not added a power calculation.

In conclusion, we agree with the Reviewer on the need to tone down our interpretative remarks and to achieve this, in this updated version of the manuscript we have referred to the study as a proof of concept investigation, which highlights the “preliminary” nature of the findings and contribution to demonstrating “feasibility” of this approach.

Throughout, the authors present data they refer to as correlation data but report r-squared rather than r-values. They also do not make it clear whether they are presenting adjusted r-squared values.  In addition Figure 3 graphs have what I would regard as regression lines-this is not appropriate for correlation analysis.  I would state that the authors need to clearly distinguish when correlation or regression analysis is being undertaken. I am satisfied that the authors undertake covariate adjustment, but this is a very risky strategy for samples that are as small as these and should be undertaken in a very limited manner.

  1. We thank the Reviewer for their comments about the clarity of which statistical test has been used. Overall, we agree with the idea that it is important to be consistent throughout the paper with the use (and the description) of the statistical methods. We have now removed all references to regression analyses. The confusion was due to the addition of the line (together with the error curves) in Figure 3, which in our view served to highlight the association between the variables on axes x and y, to the benefit of the viewer. In this updated version of the manuscript, however, this has been removed, to avoid any misunderstanding. Moreover, the choice of performing exclusively correlation analyses was also not to over-interpret the association between the neuropsychological tests and metabolic parameters, as in correlation models x and y are mathematically interchangeable and the model does not imply any statistical directionality (as a regression would do). Moreover, R-squared values have been removed and replaced with r-values. Changes are highlighted with track changes between lines 255 and 277 in the manuscript.

The authors should not state "no correlation" but "no significant correlation"

  1. We thank the Reviewer for their comment and have changed the wording as suggested on line 297, 333, 340, 349, and 362 of the revised manuscript.

The data presented is very interesting, researching AD in any of its forms is very challenging, and large samples are often difficult to obtain for studies such as this.

  1. We thank the Reviewer for their observation.

I would ask that the authors report this as a proof of concept or pilot study or stress in more detail their limitations 

  1. We are happy to make this change and have referred to the study as a” proof-of-concept report” on line 123, and 301 of the revised manuscript.

Reviewer 2 Report

My life work has been on Gly-His-Lys:Cu 2+, a human plasma constituent that declines sharply with age. It appears to strongly control or influence about 31% of the human genes. GHK-Cu does increase carbon dioxide production in live mice and isolated mouse liver slices. Gene expression studies also strongly suggest that it increases mitochondrial activity. The gene expression data is from the Broad Institute at MIT and Harvard, Cambridge, MA, USA,

Mitochondria genes stimulated by GHK        

  Probe Set ID Gene Title Fold Percent
      Change Change
1 214069_at acyl-CoA synthetase medium-chain family member 2A /// acyl-CoA synthetase medium-chain family member 2B, ACSM2A /// ACSM2B 10.92 992.06
2 220804_s_at tumor protein p73, TP73 10.38 938.37
3 215168_at translocase of inner mitochondrial membrane 17 homolog A (yeast), TIMM17A 7.89 689.04
4 218790_s_at trimethyllysine hydroxylase, epsilon, TMLHE 6.97 597.29
5 205643_s_at protein phosphatase 2, regulatory subunit B, beta, PPP2R2B 6.46 545.53
6 211196_at dihydrolipoamide branched chain transacylase E2, DBT 5.39 439.27
7 206011_at caspase 1, apoptosis-related cysteine peptidase (interleukin 1, beta, convertase), CASP1 5.32 432.07
8 222083_at glycine-N-acyltransferase, GLYAT 5.18 418.29
9 221281_at v-src sarcoma (Schmidt-Ruppin A-2) viral oncogene homolog (avian), SRC 4.99 399.39
10 207686_s_at caspase 8, apoptosis-related cysteine peptidase, CASP8 4.99 398.72
11 215977_x_at glycerol kinase, GK 4.65 365.37
12 210626_at A kinase (PRKA) anchor protein 1, AKAP1 4.15 314.84
13 222248_s_at sirtuin 4, SIRT4 4.03 302.86
14 208844_at voltage-dependent anion channel 3, VDAC3 3.87 286.55
15 222007_s_at FK506 binding protein 8, 38kDa, FKBP8 3.72 272.29
16 209961_s_at hepatocyte growth factor (hepapoietin A; scatter factor), HGF 3.69 269.31
17 217066_s_at dystrophia myotonica-protein kinase, DMPK 3.30 230.30
18 221533_at family with sequence similarity 162, member A, FAM162A 3.28 227.68
19 217294_s_at enolase 1, (alpha), ENO1 3.18 217.62
20 221284_s_at v-src sarcoma (Schmidt-Ruppin A-2) viral oncogene homolog (avian), SRC 3.16 215.98
21 220474_at solute carrier family 25 (mitochondrial oxodicarboxylate carrier), member 21, SLC25A21 3.14 214.31
22 215820_x_at sorting nexin 13, SNX13 3.02 202.28
23 203032_s_at fumarate hydratase, FH 2.85 185.30
24 211234_x_at estrogen receptor 1, ESR1 2.85 184.85
25 204515_at hydroxy-delta-5-steroid dehydrogenase, 3 beta- and steroid delta-isomerase 1, HSD3B1 2.84 183.80
26 204858_s_at thymidine phosphorylase, TYMP 2.82 181.86
27 203783_x_at polymerase (RNA) mitochondrial (DNA directed), POLRMT 2.80 179.52
28 202002_at acetyl-CoA acyltransferase 2, ACAA2 2.78 178.41
29 206957_at alanine-glyoxylate aminotransferase, AGXT 2.78 177.51
30 221037_s_at solute carrier family 25 (mitochondrial carrier; adenine nucleotide translocator), member 31, SLC25A31 2.75 175.31
31 207058_s_at parkinson protein 2, E3 ubiquitin protein ligase (parkin), PARK2 2.69 169.02
32 220047_at sirtuin 4, SIRT4 2.68 168.03
33 211789_s_at MLX interacting protein, MLXIP 2.66 165.68
34 220515_at dual specificity phosphatase 21, DUSP21 2.65 164.99
35 201159_s_at N-myristoyltransferase 1, NMT1 2.56 155.80
36 205446_s_at activating transcription factor 2, ATF2 2.56 155.64
37 201490_s_at peptidylprolyl isomerase F, PPIF 2.49 148.57
38 206068_s_at acyl-CoA dehydrogenase, long chain, ACADL 2.46 146.26
39 221941_at polyamine oxidase (exo-N4-amino), PAOX 2.46 146.21
40 217021_at cytochrome b5 type A (microsomal), CYB5A 2.42 142.18
41 206902_s_at endo/exonuclease (5'-3'), endonuclease G-like, EXOG 2.38 137.78
42 205369_x_at dihydrolipoamide branched chain transacylase E2, DBT 2.37 136.63
43 216657_at ataxin 3, ATXN3 2.36 135.95
44 214225_at protein (peptidylprolyl cis/trans isomerase) NIMA-interacting, 4 (parvulin), PIN4 2.35 134.64
45 212272_at lipin 1, LPIN1 2.35 134.57
46 210377_at acyl-CoA synthetase medium-chain family member 3, ACSM3 2.33 133.43
47 219195_at peroxisome proliferator-activated receptor gamma, coactivator 1 alpha, PPARGC1A 2.26 125.62
48 209960_at hepatocyte growth factor (hepapoietin A; scatter factor), HGF 2.25 125.00
49 214053_at v-erb-a erythroblastic leukemia viral oncogene homolog 4 (avian), ERBB4 2.25 124.87
50 206069_s_at acyl-CoA dehydrogenase, long chain, ACADL 2.22 121.89
51 203681_at isovaleryl-CoA dehydrogenase, IVD 2.20 120.21
52 217139_at voltage-dependent anion channel 1, VDAC1 2.20 120.11
53 217502_at interferon-induced protein with tetratricopeptide repeats 2, IFIT2 2.17 117.07
54 216591_s_at succinate dehydrogenase complex, subunit C, integral membrane protein, 15kDa, SDHC 2.16 115.79
55 219708_at 5',3'-nucleotidase, mitochondrial, NT5M 2.13 112.87
56 209718_at non-SMC condensin II complex, subunit H2, NCAPH2 2.12 111.75
57 210671_x_at mitogen-activated protein kinase 8, MAPK8 2.10 110.06
58 213849_s_at protein phosphatase 2, regulatory subunit B, beta, PPP2R2B 2.05 104.56
59 207472_at arginyl-tRNA synthetase 2, mitochondrial, RARS2 2.03 103.33
60 205396_at SMAD family member 3, SMAD3 2.03 102.92
61 214203_s_at proline dehydrogenase (oxidase) 1, PRODH 2.02 101.51
62        
Mitochondria genes suppressed by GHK          

  Probe Set ID Gene Title Fold Change Percent Change  
1 211577_s_at insulin-like growth factor 1 (somatomedin C), IGF1 -6.22 -521.79  
2 204309_at cytochrome P450, family 11, subfamily A, polypeptide 1, CYP11A1 -4.93 -393.16  
3 213683_at acyl-CoA synthetase long-chain family member 6, ACSL6 -4.84 -383.76  
4 204607_at 3-hydroxy-3-methylglutaryl-CoA synthase 2 (mitochondrial), HMGCS2 -4.42 -342.27  
5 207200_at ornithine carbamoyltransferase, OTC -4.31 -331.26  
6 217167_x_at glycerol kinase, GK -4.23 -323.17  
7 215352_at   -4.13 -313.34  
8 205444_at ATPase, Ca++ transporting, cardiac muscle, fast twitch 1, ATP2A1 -3.99 -299.49  
9 213830_at T cell receptor delta locus, TRD@ -3.92 -292.10  
10 213014_at mitogen-activated protein kinase 8 interacting protein 1, MAPK8IP1 -3.89 -289.06  
11 206865_at harakiri, BCL2 interacting protein (contains only BH3 domain), HRK -3.48 -247.79  
12 216540_at T cell receptor delta locus, TRD@ -3.46 -246.39  
13 206406_at sperm mitochondria-associated cysteine-rich protein, SMCP -3.07 -206.52  
14 217636_at polymerase (DNA directed), gamma, POLG -3.04 -203.68  
15 204466_s_at synuclein, alpha (non A4 component of amyloid precursor), SNCA -3.00 -200.19  
16 207101_at vesicle-associated membrane protein 1 (synaptobrevin 1), VAMP1 -2.93 -192.67  
17 219827_at uncoupling protein 3 (mitochondrial, proton carrier), UCP3 -2.92 -191.84  
18 210326_at alanine-glyoxylate aminotransferase, AGXT -2.75 -174.75  
19 211235_s_at estrogen receptor 1, ESR1 -2.61 -161.35  
20 215573_at Catalase, CAT -2.49 -149.24  
21 217497_at thymidine phosphorylase, TYMP -2.43 -142.63  
22 207968_s_at myocyte enhancer factor 2C, MEF2C -2.38 -138.18  
23 210755_at hepatocyte growth factor (hepapoietin A; scatter factor), HGF -2.38 -138.16  
24 203784_s_at DEAD (Asp-Glu-Ala-Asp) box polypeptide 28, DDX28 -2.26 -126.47  
25 204938_s_at phospholamban, PLN -2.20 -120.06  
26 215414_at Phenylalanyl-tRNA synthetase 2, mitochondrial, FARS2 -2.20 -119.82  
27 220061_at acyl-CoA synthetase medium-chain family member 5, ACSM5 -2.16 -115.57  
28 204289_at hypothetical LOC100506517, LOC100506517 -2.12 -111.53  
29 209223_at NADH dehydrogenase (ubiquinone) 1 alpha subcomplex, 2, 8kDa, NDUFA2 -2.09 -109.49  
30 207827_x_at synuclein, alpha (non A4 component of amyloid precursor), SNCA -2.03 -103.43  

GHK-Cu is used by physicians in the USA for the healing of spinal-cervical damage when injected near the injury at 0.25 mgs per day.

Russian studies report that GHK-Cu's most potent effects in rats as reductions in anxiety and pain-induced aggression.

A recent paper reported that GHK-Cu improves mental function in aged mice. 

The potential of GHK as an anti-aging peptide Yan Dou, Amanda Lee, Lida Zhu, John Morton, Warren Ladiges Aging Pathobiology and Therapeutics 2020; 2(1):58-61 58    Recent papers on GHK-Cu are  Pickart, L., & Margolina, A. (2018). Regenerative and protective actions of the GHK-Cu peptide in the light of the new gene data. International journal of molecular sciences19(7), 1987.   Pickart, L., Vasquez-Soltero, J. M., & Margolina, A. (2017). The effect of the human peptide GHK on gene expression relevant to nervous system function and cognitive decline. Brain sciences7(2), 20.   My main advice to the authors is to buy some of the sublingual GHK-Cu, 5 mgs and give it to the patients. Direct-Peptides sells it in the UK. GHK-Cu is probably safer than water.  

Author Response

Response to Reviewer 2

We thank the reviewer for the positive judgement of the paper and the authors will investigate and carefully consider GHK as a potential therapeutic strategy in our future work. As the reviewer mentions, studies have shown that GHK, by increasing gene expression, has an effect on improving mitochondrial activity and influencing cognitive decline (Pickart et al 2017). We have now mentioned in the introduction that there is evidence that GHK can influence mitochondrial activity and cognitive decline and referenced the suggested paper. This change can be found from line 113 to line 115 of the revised manuscript

Reference

Pickart, L., Vasquez-Soltero, J. M., & Margolina, A. (2017). The effect of the human peptide GHK on gene expression relevant to nervous system function and cognitive decline. Brain sciences7(2), 20.

Round 2

Reviewer 1 Report

The authors have addressed all my previous concerns